# Goniothalamin Induces Necroptosis and Anoikis in Human Invasive Breast Cancer MDA-MB-231 Cells

**DOI:** 10.3390/ijms20163953

**Published:** 2019-08-14

**Authors:** Patompong Khaw-on, Wilart Pompimon, Ratana Banjerdpongchai

**Affiliations:** 1Department of Biochemistry, Faculty of Medicine, Chiang Mai University, Chiang Mai 50200, Thailand; 2Laboratory of Natural Products, Faculty of Science and Center for Innovation in Chemistry, Lampang Rajabhat University, Lampang 52100, Thailand

**Keywords:** anoikis, breast cancer, epithelial-mesenchymal transition, goniothalamin, necroptosis, pan-caspase inhibitor

## Abstract

Goniothalamin (GTN) is toxic to several types of cancer cells in vitro. However, its effects on non-apoptotic cell death induction of human cancer cells have been poorly documented. Here, an investigation of the anti-cancer activity of GTN and the molecular signaling pathways of non-apoptotic cell death in the invasive human breast cancer MDA-MB-231 cell line were undertaken. Apoptotic cell death was suppressed by using a pan-caspase inhibitor (Benzyloxycarbonyl-Val-Ala-Asp-[O-methyl]-fluoromethylketone), z-VAD-fmk) as a model to study whether GTN induced caspase-independent cell death. In the anoikis study, MDA-MB-231 cells were cultured on poly-(2-hydroxyethyl methacrylate)- or poly-HEMA- coated plates to mimic anoikis-resistance growth and determine whether GTN induced cell death and the mechanisms involved. GTN and z-VAD-fmk induced human breast cancer MDA-MB-231 cells to undergo necroptosis via endoplasmic reticulum (ER) and oxidative stresses, with increased expressions of necroptotic genes such as *rip1*, *rip3*, and *mlkl*. GTN induced MDA-MB-231 cells to undergo anoikis via reversed epithelial-mesenchymal transition (EMT) protein expressions, inhibited the EGFR/FAK/Src survival signaling pathway, and decreased matrix metalloproteinase secretion.

## 1. Introduction

Goniothalamin (GTN) is a phytosteryl-lactone found in the genus *Goniothalamus* and has been widely used as a folk medicine in Southeast Asia. GTN inhibits the proliferation of HCG-27, MCF7, PAN C-1, and HeLa cancerous cell-lines and noncancerous fibroblast NIH3T3 cell lines cultured in vitro [1], and also induces apoptosis in Jurkat T-cells and human promyelocytic leukemia HL-60 cells via caspase-3, -7 and poly(ADP-ribose)polymerase-1 (PARP-1) cleavage [2,3]. GTN causes cell cycle arrest at the Gap2/Mitosis (G_2_/M) phase in human breast cancer MDA-MB-231 cells [4] and induces deoxyribonucleic acid (DNA) damage and ROS production, which subsequently lead to apoptosis in many cell lines [5,6]. GTN also induces necrosis in MCF-7 cells [7] through mechanisms that are still not completely understood.

Necroptosis, a programmed form of necrosis, is regulated through caspase-independent cell death mechanisms. Death receptors including tumor necrosis factor receptor (TNFR), Fas (a death receptor binds to Fas ligand), and tumor necrosis factor-relating apoptosis-induced ligand-receptor (TRAIL-R) also activate necroptosis by recruiting necrosome formation consisting of receptor-interacting serine/threonine protein kinases 1 (RIP1) and RIP3 [8,9]. Mixed-lineage kinase domain-like (MLKL) protein, a substrate of necrosome, translocates to plasma membrane forming channels, which leads to the influx of Ca^2+^ ions and subsequently necroptotic cell death [10]. Oxidative stress also plays a critical role in intrinsic necrosis mechanisms. Moreover, alkylating DNA-damage agents trigger caspase-independent necroptosis, which involves the serial activation of various proteins and enzymes including PARP-1, calpains, Bcl-2 associated X protein (Bax), and apoptosis-inducing factor (AIF) as crucial regulators of a rapid increase in inner mitochondrial membrane permeability and caspase-independent necroptosis [11,12].

Anoikis is a form of programmed cell death and regulates cell versus extracellular matrix (ECM) interaction and anchorage-independent growth factor receptor signaling, which play pivotal roles in cancer colonization and metastasis. Some types of cancers, for example, breast cancer MDA-MB-231 cells, survive without ECM contact and grow in anchorage-independent milieus. Anoikis resistance is characterized by metastasized potential and invasive character [13]. Moreover, growth factor signaling pathways such as EGFR, Src, and ERK also play a crucial role in anoikis resistance as well as integrin type alteration [14]. Apoptosis-related proteins, for example, Fas, Bax, Bcl-xL, and Bim-EL, also influence the sensitization of resistant cancer cells to undergo anoikis [14].

Invasive breast cancer evades apoptotic cell death and becomes drug-resistant and metastasizes to other organs [15]. This study aimed to characterize different signaling pathways of invasive breast cancer MDA-MB-231 cell necroptosis and anoikis-sensitizing effects induced by GTN. The results indicated that GTN induced necroptosis in caspase-inhibited MDA-MB-231 cells through oxidative stress, high intracellular Calcium levels, and necroptotic molecules such as calpain, RIP1, RIP3, MLKL, and AIF. GTN also reversed anoikis-resistant MDA-MB-231 cells, rendering them sensitive to anoikis induction through decreased levels of EGFR, FAK, Src proteins, and epithelial-mesenchymal transition (EMT) alteration, which normally functions in survival pathways. This study investigated the properties of GTN as a natural product of non-apoptotic cell death via necroptosis and anoikis as novel signaling pathways. Necroptosis can function as a reciprocal backup mechanism of apoptosis [16]; therefore, the examination of natural products-induced necroptosis is important to discover new mechanisms for cancer treatment and chemoprevention. Novel pathways to prevent cancer cell migration and metastasis are also beneficial by anoikis re-sensitization.

## 2. Results

### 2.1. Cytotoxic and Necroptotic Effects on MDA-MB-231 Cells Induced by GTN and z-VAD-fmk Co-Treatment

The cytotoxic effect and caspase-independent cell death of GTN-treated MDA-MB-231 cells were determined. GTN exhibited toxicity against MDA-MB-231 cells under co-treatment with 10 µM of pan-caspase inhibitor (z-VAD-fmk) [17] (Figure 1A). The inhibitory concentration at 50 percent (IC_50_) was 33.82 µM compared to 44.65 µM without z-VAD-fmk, but GTN was less cytotoxic to MCF-10A, which is a non-tumorigenic human breast epithelial cell line with an IC_50_ of >80 µM. The apoptosis assay revealed that GTN combined with z-VAD-fmk increased the PI-positive cell population significantly when compared to GTN alone (Figure 1B). Transmission electron micrographs (TEM) revealed the cell morphology of untreated-MDA-MB231 cells which was an intact and sealed cell membrane (Figure 1C, green arrowhead), the same as z-VAD-fmk treated cells that are not toxic toward MDA-MB-231 cells (Figure 1F, green arrowhead). GTN and z-VAD-fmk co-treatment demonstrated the MDA-MB-231 cell loss of their cell membrane integrity (red arrowhead as in Figure 1D) as well as that of the positive control hydrogen peroxide (Figure 1E, red arrowhead). In contrast, the GTN-treated (alone) cells revealed cell shrinkage and DNA condensation and membrane blebbing (blue arrowhead), which is an apoptosis feature (Figure 1G). When combined with our previous study, it revealed that GTN alone induced MDA-MB231 cell apoptosis [18]. Hence, z-VAD-fmk may shift the mode of cell death induced by GTN to be necrosis rather than apoptosis, which is mediated through caspases, therefore apoptosis was inhibited by the pan-caspase inhibitor.

### 2.2. GTN Co-Treated with z-VAD-fmk Induced Caspase-Independent Necroptosis but Mediated through Calpain Activity

Comparison of GTN with z-VAD-fmk treatment or without treatment showed that it did not induce caspase-3, -8, and -9 activities significantly, as shown in Figure 2A. The non-caspase enzyme, calpain, also plays an important role in non-apoptotic cell death [19]. GTN with z-VAD-fmk co-treatment significantly increased calpain activity when compared to the control and GTN treatment alone (Figure 2B). This combined GTN plus z-VAD-fmk effect was inhibited by the calpain inhibitor (z-LLY-fmk) with the same character as the condition of co-treatment of the positive control (the active calpain I) with the calpain inhibitor (z-LLY-fmk) (Figure 2B).

### 2.3. Induction of High Cellular ROS Production and Increased Cytosolic Calcium Level by GTN Co-Treated with z-VAD-fmk in MDA-MB-231 Cells

Previous studies have indicated that necroptosis is associated with cellular ROS generation and cytosolic Ca^2+^ level alterations [20]. To study whether GTN and z-VAD-fmk induced-necroptosis involved free radical generation and high Ca^2+^ level alteration, GTN was co-treated with z-VAD-fmk, where the superoxide anion radicals and peroxide radicals increased when measured by dihydroethidium (DHE) and 2′,7′–dichlorodihydrofluorescin diacetate (DCFH-DA), respectively, indicating that both ROS accelerated significantly in the GTN and z-VAD-fmk co-treated-cancer cells. GTN treatment alone also increased the ROS levels in MDA-MB-231 cells, although the significance level was lower than GTN with the pan-caspase inhibitor. All these effects were inhibited by countered treatment with *N*-acetylcysteine (NAC), an antioxidant (Figure 3A,B), respectively.

The cytosolic Ca^2+^ level also increased as Fluo3 AM fluorescence intensity in the GTN and z-VAD-fmk co-treated cells, indicating that Ca^2+^ and ROS induced endoplasmic reticulum (ER) and oxidative stresses, respectively, leading to necroptotic cell death (Figure 3C). GTN-treated cells in the absence of z-VAD-fmk also allowed apoptosis to occur with an increased cytosolic Ca^2+^ level (data not shown), indicating that GTN induced apoptosis and necroptosis via an endoplasmic recticulum (ER)-stress pathway.

### 2.4. Signaling Pathway of GTN-Induced Necroptosis When Co-Treated with z-VAD-fmk

To investigate the role of necroptosis molecular alteration in the signaling pathways of GTN and pan-caspase inhibitor co-treatment, GTN and z-VAD-fmk-treated MDA-MB-231 cells were investigated by Western blotting for RIP3 expression. GTN and the pan-caspase inhibitor treatment showed an increase of RIP3 protein level in a dose-dependent manner. GTN treatment alone did not induce RIP3 protein expression, which confirmed that necroptosis was exclusively induced in the GTN plus z-VAD-fmk-treated cells (Figure 4A,B). Real-time RT-PCR for genes expressions responsible for necroptosis, including *rip1*, *rip3*, and *mlkl* gene expressions also significantly increased in a dose-dependent manner (Figure 4C–E).

Necroptosis induced by ROS and calpain truncates the apoptosis-inducing factor (AIF) and leads to chromatolysis and necroptosis [21]. The truncated AIF level was investigated by Western blotting. In the GTN and z-VAD-fmk co-treated MDA-MB-231 cells, the truncated AIF level increased as demonstrated in Figure 4A,B. AIF is also released from mitochondria in the apoptosis pathway and is responsible for the loss of mitochondrial transmembrane permeability and plays a role in nuclear chromatin condensation induction in apoptosis [22]. GTN-treated cells without z-VAD-fmk also increased the AIF level compared to the control, but less than the GTN and z-VAD-fmk co-treatment at the same doses (Figure 4A,B) due to the caspase-independent pathway.

### 2.5. Goniothalamin-Reversed Anoikis Resistance in Human Breast Cancer MDA-MB-231 Cell Death via Inhibition of EGFR/FAK/Src and EMT Pathways

Anoikis is another form of apoptotic cell death that is an inappropriate cell response toward ECM interaction as loss of cell to cell adhesion [23]. Invasive human breast cancer cells are usually resistant to chemotherapeutic treatment and develop anoikis resistance [14]. The effects of GTN on invasive breast cancer MDA-MB-231 anoikis resistance were investigated and clarified for the signaling pathways. Breast cancer MDA-MB-231 cells were cultured in pre-coated poly-HEMA plates for 48 h to induce the spheroid formation of the cancer cells to mimic the anoikis-resistant (non-adherent) condition [24]. Spheroid MDA-MB-231 cells were treated with GTN at various concentrations for 24 h. Cell viability of spheroid MDA-MB-231 cells after GTN treatment decreased in a dose-dependent manner, with inhibitory concentrations of twenty (IC_20_) and fifty percent (IC_50_) at 18 and 58 µM, respectively (Figure 5A).

In breast cancer, collagen type I deposits nearly 10-fold stiffening of the mammary stroma. Cancer invasion overcoming these dense matrices is related to matrix-metalloproteinase (MMP)-mediated ECM degradation that migrates to a secondary site [25]. Matrix metalloprotease (MMP) activity was measured by zymography assays. Un-treated adherent breast cancer cells secreted MMP-9 higher than the detached-culture cells at a basal level. In contrast, MMP-2 was secreted in an anchorage-independent environment. Our data demonstrate that GTN-treated detached MDA-MB-231 cells decreased MMP-9 and MMP-2 in a time-dependent manner with 20 percent inhibitory growth concentration (Figure 5B–E).

Anoikis resistance and epithelial-mesenchymal transition (EMT) are considered prerequisites for cancer cells to metastasize [14]. The matrix metalloproteinase-9 catalyzes and degrades the ectodomain of E-cadherin, disrupting junctional integrity in cancer cells and contributing to EMT [26]. Spheroid MDA-MB-231 cells exhibited alteration of EMT markers, including increased levels of vimentin and N-cadherin; however, the E-cadherin level decreased when compared with adherent MDA-MB-231 cells (Figure 5F,G).

Whereas in GTN-treated spheroid breast cancer MDA-MB-231 cells, EMT markers such as vimentin and N-cadherins were significantly downregulated, but the expression of E-cadherin was enhanced when compared to adherent breast cancer cells (Figure 5F,G). GTN also inhibited the expression of survival signaling proteins, viz. EGFR, FAK, and Src while the expression of the pro-apoptotic BH3-only protein, Bim-EL, a marker of anoikis, increased in a dose-dependent manner. Conclusively, GTN induced anoikis sensitivity in the anoikis-resistant MDA-MB-231 cell model via Bim-EL, EMT, and survival protein alteration pathways (Figure 5F,G).

## 3. Discussion

GTN has been reported to induce apoptosis cell death in many cancer cell types [1]. Cisplatin also shows the potential of necroptosis induction when co-treated with pan-caspase inhibitor on gastric cancer [27]. Reports regarding other natural products or extracts, namely green tea extract, shikonin, and columbianadin, demonstrate the results of necroptosis induction, which is called “regulated necrosis” [28,29,30]. Necroptotic cell death can be combined or used alone with greater probability and benefits for cancer therapy when considering the mechanism of action at the molecular level. However, investigation of GTN on other programmed cell death inductions and the mechanisms involved remain unclear. Necrosis morphology has previously been found in MCF-7 breast cancer cells [7] by TEM. Here, we revealed the different modes of cell death in GTN-treated MDA-MB-231 cells treated with z-VAD-fmk. An ultra-fine structure of GTN co-treated with z-VAD-fmk cells exhibited large vacuoles, ruptured cell membranes, and released cellular organelles, which were the same as hydrogen peroxide-induced necrosis positive control (Figure 1). This indicates that GTN induced cell death in different modes by shifting from apoptosis to necrosis when co-treated with the pan-caspase inhibitor. Regarding the cytotoxicity of normal cells, goniothalamin was less toxic to normal cell types when compared to cancer cell types [31,32,33], in addition, this study also added the cytotoxicity data of the normal mammalian gland epithelial MCF-10A cell line, of which GTN was less cytotoxic at ten percent when compared to the control cells (Figure 1). This indicates that goniothalamin is a natural compound, which possesses specific toxicity toward cancer cells.

Intracellular ROS and cytosolic calcium ion levels were involved in the initial necroptosis-induced GTN co-treated with z-VAD-fmk cells. Previous studies reported that only hydrogen peroxide was produced in GTN-treated cancer cells [4,34]. Our data demonstrated that the superoxide anion radical level also increased, which then generated hydrogen peroxide radicals [35]. From this study, the elevation of ROS was raised in a couple of hours and declined to the baseline after 12 h of treatment (Appendix A). Punganuru et al. reported that R-goniothalamin induced ROS generation at three hours of treatment in SKBR3 breast cancer cells harboring an R175H mutant p53 and that reactive oxygen species (ROS) were subsequently conjugated with glutathione making the ROS level decline [36].

The cytosolic calcium level was related to intracellular free radicals [37]. Our research demonstrated that accelerated cytosolic calcium ion levels triggered ER-stress and downstream signaling molecules as necroptosis inducers. Calpain, a calcium-dependent cysteine protease, was induced in GTN-induced necroptosis by the pan-caspase inhibitor (z-VAD-fmk) model. Calpain also serves as a necroptosis initiator, responsible for AIF cleavage or proteolysis, which leads to chromatolysis and RIP1 induction via JNK activation [38]. GTN co-treated with z-VAD-fmk demonstrates increased expressions of truncated AIF, which is responsible for a high level of calpain activity. Apoptosis-inducing factor (AIF) is not only a pro-apoptosis protein but is also responsible for caspase-independent necroptosis modulator [21]. Additionally, AIF has become a novel molecule, which exhibits the interconnection between apoptosis and other programmed cell deaths [39]. Moreover, goniothalamin induces necroptosis via the RIP3-independent mechanism, which is an alternative pathway for the molecular signaling of necroptosis.

RIP1 is an essential protein in the death-inducing signaling complex (DISC) involved in death-receptor-initiated cell death, apoptosis, and necroptosis [40]. The receptor-specific necroptosis signaling, RIP3 pathway is a crucial upstream kinase that mediates downstream signaling such as MLKL [41]. Here, we also demonstrated the potential of GTN to induce the breast cancer-derived cell line MDA-MB-231 to undergo necroptosis via the RIP3 pathway with increased RIP3 protein expression, whereas the *rip1*, *rip3*, and *mlkl* gene expressions were also significantly enhanced, finally leading to necroptosis as illustrated in Figure 6. Interconnections of the signaling pathways that induce necroptosis by GTN and z-VAD-fmk co-treatment and the crosstalk between the extrinsic RIP3 pathway and the intrinsic AIF pathway requires further elucidation.

GTN-induced apoptosis associated with other programmed cell deaths and autophagy involve**s** the induction of the JNK1 pathway and inhibition of the ERK/Akt survival pathways [42]. GTN inhibits matrix metalloproteinase MMP-2 and MMP-9 activities, which are responsible for cell migration and metastasis [43]. We also found that GTN attenuated MMP-2 secretion in both adherent and suspension (spheroid) MDA-MB-231 cell culture, which could reduce the potential of breast cancer cell migration and metastasis. Anoikis is a form of programmed cell death induced by detachment from the ECM [13]. Invasive breast cancer cells become anoikis-resistant, resulting in cell migration and metastasis to other organs [13]. The EGFR/FAK/Src pathway is crucial for signaling angiogenesis, cell survival, and anoikis resistance in various types of cancer cells [44]. Our results exhibited the first and beneficial role of GTN in the death sensitization of anoikis-resistant MDA-MB-231 cells by inhibiting survival signaling via the EGFR, FAK, and Src pathways. Additionally, epithelial to mesenchymal transition (EMT) proteins play essential roles in invasion and metastasis, which are consequences of anoikis resistance in cancer cells [45]. The present study is the first report to demonstrate that GTN reversed EMT protein expressions of vimentin, N-cadherins, and E-cadherins to adherent expression pattern levels, and the cancer cells came to have an adherent character of epithelial but not mesenchymal cells in a dose-response manner, as illustrated in Figure 6. However, the key molecules that initiated anoikis cell death by GTN still require further investigation to illustrate how they affect EMT proteins and/or inhibit survival pathways.

## 4. Materials and Methods

### 4.1. Plant Material and Chemicals

The leaves and twigs of *Goniothalamus griffithii* were collected in January 2011 from the Chiang Mai Province of Thailand and identified by the Forest Herbarium, Department of National Park, Wildlife and Plant Conservation, Ministry of Natural Resources and Environment, Bangkok, Thailand, where a voucher specimen (BKF16447) has been deposited. GTN was extracted, purified, and identified by Professor Wilart Pompimon, Lampang Rajabpat University, Lampang, Thailand. The active compound, goniothalamin, was isolated from the dried and powdered leaves and twigs of *G. griffithii* by using ethyl acetate followed by hexane. The crude extract was subjected to column chromatography. Then the subfraction was recrystallized from chloroform-ethanol for purification. The goniothalamin identification data and its molecular structure were elucidated by spectroscopy techniques viz. NMR, IR, and UV. The spectral data were the same as in a previous report [46].

A stock solution of GTN was prepared in dimethyl sulfoxide (DMSO) at a concentration of 20 mM and diluted in Dulbecco’s Modified Eagle’s Medium (DMEM) (Gibco, Carlsbad, CA, USA) in each experiment. A pan-caspase inhibitor (z-VAD-fmk) (Abcam, Cambridge, UK) was reconstituted in DMSO and used at 10 µM of the final concentration to inhibit caspase activity. The maximum concentration of DMSO used in all experiments did not exceed 0.8 percent to avoid the toxic effect of vehicle control.

### 4.2. Cell Culture

Non-tumorigenic breast epithelial cells, MCF-10A, were purchased from the American Type Culture Collection (ATCC^®^, Manassas, VA, USA) (Number: CRL-10317™, Lot Number: 64066742). MCF-10A cells were cultured in DMEM: Ham’s F-12 (1:1) supplemented with 100 ng/mL cholera toxin, 20 ng/mL epidermal growth factor (EGF), 0.01 mg/mL insulin, 500 ng/mL hydrocortisone, and 5% sterile-filtered horse serum. The cell line was incubated at 37 °C under 5% CO_2_ atmosphere. The human breast cancer MDA-MB-231 cell line was a gift from Prof. Prachya Kongtawelert, Excellence Center for Tissue Engineering and Stem Cell, Department of Biochemistry, Faculty of Medicine, Chiang Mai University, Chiang Mai, Thailand. MDA-MB-231 cells were cultured in DMEM supplemented with 10% of inactivated fetal bovine serum (FBS, 10%), penicillin (1 × 10^6^ U/L), and streptomycin (100 mg/L) at 37 °C in a humidified atmosphere of 5% CO_2_. For the anoikis resistance model, MDA-MB-231 cancer cells were plated on poly (2-hydroxyethyl methacrylate)-(Sigma-Aldrich, St. Louis, MO, USA), poly-HEMA-, coated plate to prevent cell adhesion. After 80% confluence, the adherent cells were trypsinized with 0.05% trypsin (Thermo Fisher Scientific, Waltham, MA, USA).

### 4.3. MTT (3-(4,5 dimethylthiazol-2yl)-2,5 diphenyltetrazolium bromide) Assay

MDA-MB-231 cells were seeded in 96-well culture plates at a concentration of 1 × 10^4^ cells/well, then treated with 10 µM z-VAD-fmk and co-treated with various concentrations of GTN for 24 h. To investigate anoikis, MDA-MB-231 cells were seeded on poly-HEMA-coated plates and treated with GTN at various concentrations. Cell viability and inhibitory concentration (IC) values were determined by MTT dye using spectrophotometry and compared to untreated cells. PBMCs were used as normal cell control, and inhibitory concentrations at 20, and 50 percent were used as treatment concentrations in each experiment.

### 4.4. Determination of Caspases-3, -8, -9 and Calpain Enzyme Activities

Caspase activity was determined by Colorimetric Protease Assay Kits (Invitrogen, Thermo Fisher Scientific Inc., MA, USA), whereas calpain activity was measured by Fluorometric Assay Kit (ab65308, Abcam, Cambridge, UK). Briefly, the IC_50_ of GTN and z-VAD-fmk-treated cell pellets were lysed with RIPA lysis buffer on ice, and an equal amount of proteins from each cell pellet was prepared. Caspase-8 (Ile-Glu-Thr-Asp, IETD)-*p*NA), caspase-3 (Asp-Glu-Val-Asp, DEVD-*p*NA), caspase-9 (Leu-Glu-His-Asp, LEHD-*p*NA) chromogenic substrates and calpain ac-Leu-Leu-Tyr-7-amino-trifluoromethyl coumarin (ac-LLY-AFC) fluorogenic substrate were added for 1 h at 37 °C. Caspase activity as optical density was measured at a wavelength of 405 nm by a spectrophotometric microplate reader. Calpain activity was determined by a Synergy™ H4 fluorescence microplate reader (BioTek, Winooski, VT, USA) at a wavelength of 400/505 nm (excitation/emission).

### 4.5. Transmission Electron Microscopy (TEM)

Human breast cancer MDA-MB-231 cells were seeded and treated with IC_50_ concentrations of GTN and z-VAD-fmk combinations for 24 h. Briefly, after treatment, the cells were fixed in 2% osmium tetroxide before resin-embedding. Cell morphology was examined under a transmission electron microscope and compared to MDA-MB-231 cells treated with 0.03% hydrogen peroxide as a positive necrosis control.

### 4.6. Apoptosis Determination by Annexin V-FITC/PI Staining Employing Flow Cytometry

After treatment with z-VAD-fmk and GTN at inhibitory concentrations of twenty-percent (IC_20_) and fifty-percent (IC_50_) for 24 h, cell suspensions were stained with annexin V-FITC and PI with the Annexin Fluos Kit (Roche Applied Science, Penzberg, Germany) for 15 min and then processed through a flow cytometer. The PI-positive population was analyzed as necrotic cells by BD FACSDIVA™ (BD Biosciences, Franklin Lakes, NJ, USA).

### 4.7. Assessment of Cytosolic Calcium Ion Level

MDA-MB-231 cells were treated with z-VAD-fmk and GTN at IC_20_ and IC_50_ for 24 h and then 10 μM Fluo3 AM were added (Thermo Fisher Scientific, Waltham, MA, USA) to measure cytosolic Ca^2+^ ion levels before incubation at 37 °C for 15 min. After washing twice with PBS, the cells were analyzed by a flow cytometer.

### 4.8. Production of Intracellular Reactive Oxygen Species (ROS) Determined by DCFH-DA and DHE Assays

MDA-MB-231 cells were simultaneously treated with GTN at various concentrations and combined with or without 10 µM of z-VAD-fmk for an hour. The cells were then added to 40 μM 2’,7’-dichlorodihydrofluorescein diacetate (DCFH-DA) and 20 μM dihydroethidium (DHE) to detect hydrogen peroxide and superoxide anion radicals, respectively. After 30 min of incubation, fluorescence intensity was measured by a Synergy™ H4 Fluorescence Microplate Reader (BioTek, Winooski, VT, USA) at excitation/emission wavelengths 485/530 nm and 535/635 nm for DCF and DHE, respectively.

### 4.9. Immunoblotting Analysis

MDA-MB-231 cells were treated with GTN at various concentrations with or without 10 µM of z-VAD-fmk for 24 h. Protein lysates were extracted by RIPA supplemented with protease inhibitor cocktail tablets (Roche Applied Science, Penzberg, Germany). Immunoblotting was performed as previously described [47]. Briefly, total protein extracts were separated by 12% SDS-PAGE and then transferred to a nitrocellulose membrane. Specific antibodies against RIP3, AIF, vimentin, N-cadherins, E-cadherins, EGFR, FAK, Src, and β-actin (Abcam, Cambridge, UK) were probed to each membrane. After incubation with a secondary antibody tagged with horseradish peroxidase (HRP) (Abcam, Cambridge, UK), enhanced chemiluminescence (Pierce™, Thermo Fisher Scientific, MA, USA) was developed and exposed to x-ray film. Band density was measured by ImageJ software (version 1.52o 23 April 2019, National Institutes of Health (NIH), Rockville, MD, USA ) and reported as folds of control relative level; each band was normalized by β-actin for loading control.

### 4.10. Gelatin Zymography

Supernatants were collected from cell cultures treated with GTN at twenty percent inhibitory concentration for 12 and 24 h. A total of 50 µg of total supernatant proteins were resolved on non-reducing SDS-PAGE using 8% polyacrylamide gels containing 0.1% SDS and 1% gelatin. After electrophoresis, the gels were washed with 2.5% Triton-X100 for 30 min. The gel was transferred to a developing buffer (50 mM Tris containing 0.2 M NaCl, 5 mM CaC1_2_, 1 μM ZnCl, 0.02% NaN_3_) at room temperature for 30 min, and then the gels were replaced with a fresh developing buffer and incubated at 37 °C for 18 h. Coomassie Brilliant Blue R-250 staining was used to reveal gelatin-clear zones created by MMPs. Band intensity was measured by ImageJ software (version 1.52o 23 April 2019, National Institutes of Health (NIH), Rockville, MD, USA) and reported as folds of control relative level.

### 4.11. Quantitative Real-Time Reverse Transcription Polymerase Chain Reaction (Real-Time RT-PCR)

After various concentrations of GTN treatment combined with z-VAD-fmk (10 µM) in MDA-MB-231 cells for 24 h, RNA was isolated from the cell pellets using an Illustra RNAspin Mini Kit (GE Healthcare, Little Chalfont, UK). Total RNAs were reversed to obtain complementary DNA (cDNA) using a Tetro cDNA Synthesis Kit (Bioline Reagents Ltd., MA, USA). The following primers were used: *rip3* forward primer: 5′-TGCTGGAAGAGAAGTTGAGTTGC-3′; *rip3* reverse primer: 5′-CTGTTGCACACTGCTTCGTACAC-3′:*rip1* forward primer: 5′-GGCATTGAAGAAAAATTTAGGC -3′; *rip1* reverse primer: 5′-TCACAACTGCATTTTCGTTTG-3′: *mlkl* forward primer: 5′-AGAGCTCCAGTGGCCATAAA-3′; *mlkl* reverse primer: 5′-TACGCAGGATGTTGGGAGAT-3′: *gapdh* forward primer: 5′-TGCACCACCAACTGCTTAGC-3′; *gapdh* reverse primer: 5′-GGCATGGACTGTGGTCATGAG-3′ (Integrated DNA Technologies, Coralville, IA, USA). Quantitative real-time RT-PCR assays were performed with the SensiFAST™ SYBR^®^ Lo-ROX Kit (Bioline Reagents Ltd., MA, USA) and the QuantStudio™ 6 Flex Real-Time PCR System (Thermo Fisher Scientific, Waltham, MA, USA). The *gapdh* gene was used to normalize all data.

### 4.12. Statistical Analysis

Data were presented as the mean ± standard deviation (S.D.) from triplicates of three independent experiments using one-way analysis of variance (ANOVA), with a comparison between two groups of data analyzed by the Tukey’s test. Statistical significance was considered when * *p* < 0.05, or ** *p* < 0.01 when compared to the control and # *p* < 0.05 or ## *p* < 0.01 compared to GTN treatment alone.

## 5. Conclusions

Here, the non-apoptotic cell death mechanisms of necroptosis and anoikis induced by GTN in human breast cancer MDA-MB-231 cells were investigated. The presence of the pan-caspase inhibitor (z-VAD-fmk) mimicked apoptosis inhibition, which occurs in many types of invasive cancer cells. GTN exhibited high potency to induce cell death via expressions of *rip1*, *rip3*, and *mlkl* genes as an alternative pathway of programmed necroptosis. ER, and reactive oxygen species-mediated GTN induced necroptosis in caspase-abrogated breast cancer cells. Furthermore, GTN inhibited metastasized breast cancer cells (in the detachment model) by inducing anoikis via decreased MMP secretion, a reversal of the EMT proteins, and attenuated EGFR/FAK/Src survival pathways. Taken together, GTN strongly demonstrated anti-cancer effects as apoptosis-resistance by necroptosis induction, and in anchorage-independent breast cancer cells by anoikis re-sensitization.

## Figures and Tables

**Figure 1 ijms-20-03953-f001:**
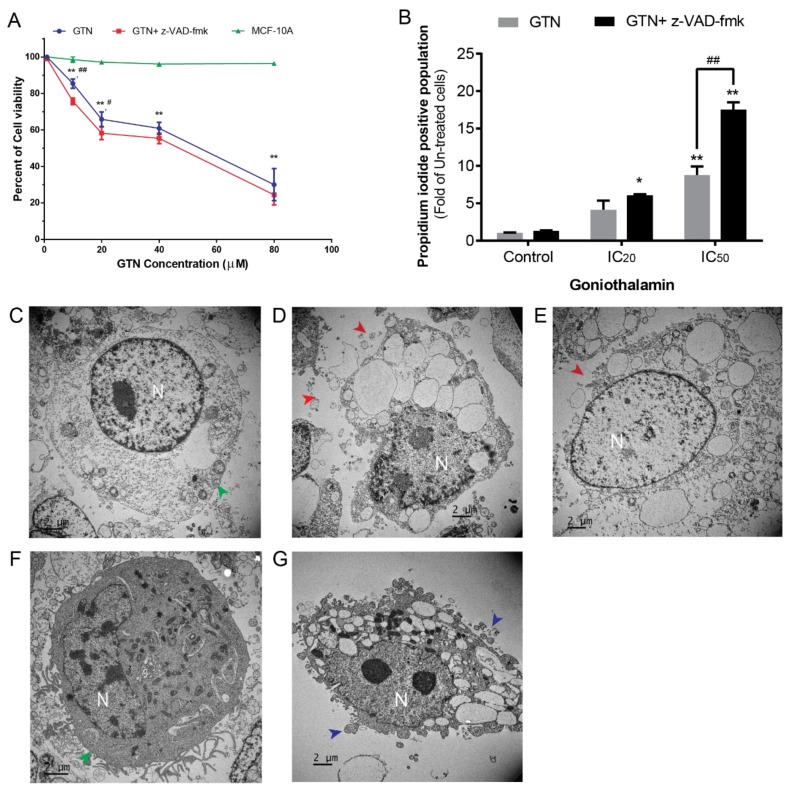
The cytotoxic effect and necroptosis induction of GTN on MDA-MB-231 cells. The cell viability of goniothalamin (GTN) in the presence or absence of z-VAD-fmk against human breast cancer MDA-MB-231 cells and MCF-10A cells were determined by the MTT ((3-(4,5-dimethylthiazol-2-yl)-2,5-diphenyltetrazolium bromide) assay (**A**). Percent of propidium iodide-positive cells in the condition with or without z-VAD-fmk at a GTN concentration of IC_0_, IC_20_, and IC_50_ were measured by annexin V-fluorescein isothiocyanate and PI staining followed by FACs (**B**). Transmission electron micrograph of the untreated MDA-MB-231 cell (**C**) compared to GTN co-treated with z-VAD-fmk (**D**), 0.03% hydrogen peroxide-treated cell as a positive control (**E**), z-VAD-fmk-treated cell (**F**) and GTN-treated cell (**G**) (N = nucleus, green arrowhead = intact membrane, red arrowhead = collapsed membrane and blue arrowhead = blebbing membrane). The results were collected from three independent experiments in triplicate, the significance of statistical values compared to the control (without treatment) are marked with * *p* < 0.05; ** *p* < 0.01, and compared to GTN alone are marked with # *p* < 0.05; ## *p* < 0.01.

**Figure 2 ijms-20-03953-f002:**
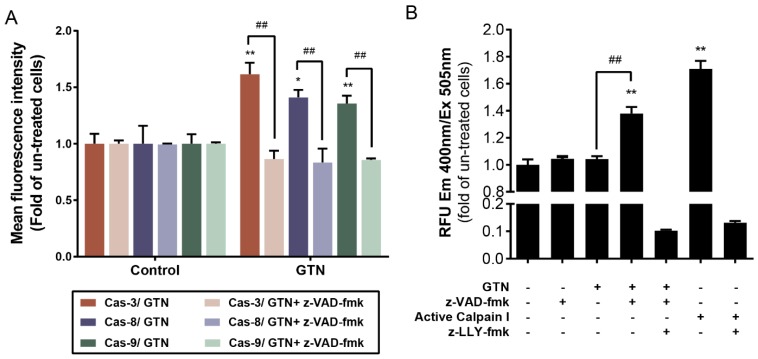
Caspase-independent MDA-MB-231 necroptosis but via calpain enzyme activity. Caspase-3, -8, and -9 activities in the presence of GTN at an inhibitory concentration of fifty-percent with or without z-VAD-fmk co-treatment compared with the z-VAD-fmk treatment alone as control (**A**). Calpain activity increased in GTN plus z-VAD-fmk, which was measured by fluorescence intensity, but the effect was inhibited by a pharmacological calpain tripeptide inhibitor (z-LLY-fmk) that confirmed the mode of programmed cell death to be necroptosis (**B**). Data were obtained from three independent experiments in triplicate, the significance of statistical values when compared to the control (without treatment) are marked with * *p* < 0.05 or ** *p* < 0.01 and compared to GTN alone is marked with ## *p* < 0.01.

**Figure 3 ijms-20-03953-f003:**
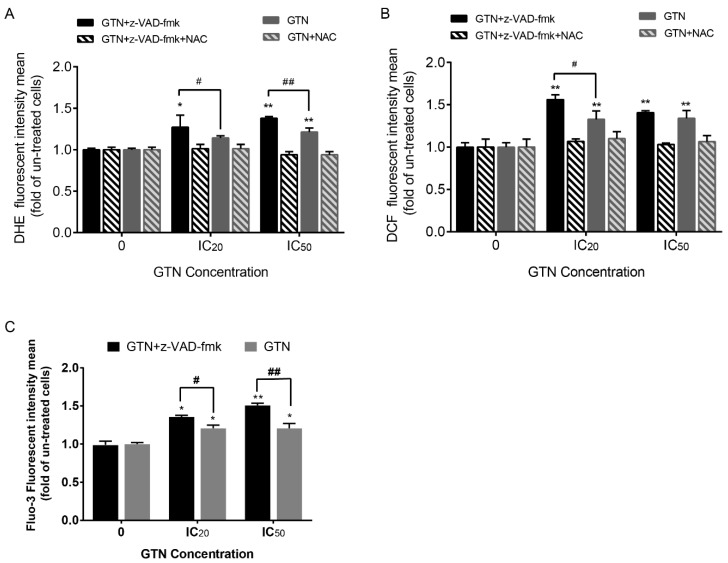
Endoplasmic reticulum (ER) and oxidative stress involvement in GTN and z-VAD-fmk-induced MDA-MB-231 necroptosis. ROS production was measured by dihydroethidium (DHE) (**A**), and 2’,7’-dichlorodihydrofluorescin diacetate (DCFH-DA) (**B**) represents superoxide anion radicals and hydrogen peroxide radicals, respectively, followed by fluorescent intensity detection. *N*-acetylcysteine (NAC) is an antioxidant, which was used to inhibit intracellular ROS production. Cytosolic calcium ion level in GTN-treated cells in the presence or absence of z-VAD-fmk, which was stained by Fluo3-AM dye and fluorescence intensity was measured by FACS (**C**). From three independent experiments in triplicate, the significance of the statistical values compared to the un-treatment group are marked with * *p* < 0.05; ** *p* < 0.01, and compared to GTN alone as # *p* < 0.05 and ## *p* < 0.01.

**Figure 4 ijms-20-03953-f004:**
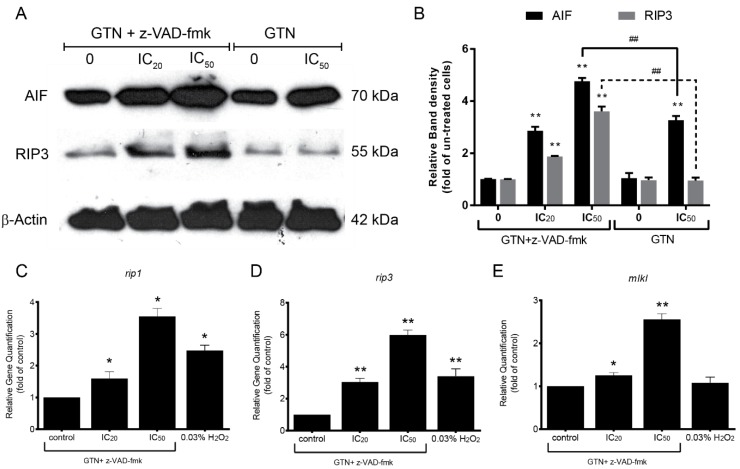
Enhanced necroptosis protein and gene expressions in the GTN and pan-caspase inhibitor-treated MDA-MB-231 cells. The Apoptosis-inducing factor (AIF) and RIP3 protein band density representing protein expression were performed by Western blotting (**A**) and the average band density compared to β-actin, a loading control, was measured by ImageJ software, and the number of folds compared to β-actin are illustrated in the bar graph (**B**). Bar graphs of *rip1, rip3*, and *mlkl* mRNA expressions are demonstrated and normalized with *gapdh* as a housekeeping gene. The gene expression was performed by SYBR green-based real-time RT-PCR (**C**, **D,** and **E**). The data were obtained from three independent experiments in triplicate, the significance of statistical values compared to the z-VAD-fmk-treated alone group are marked with * *p* < 0.05; ** *p* < 0.01 and compared to the GTN alone group are marked with ## *p* < 0.01.

**Figure 5 ijms-20-03953-f005:**
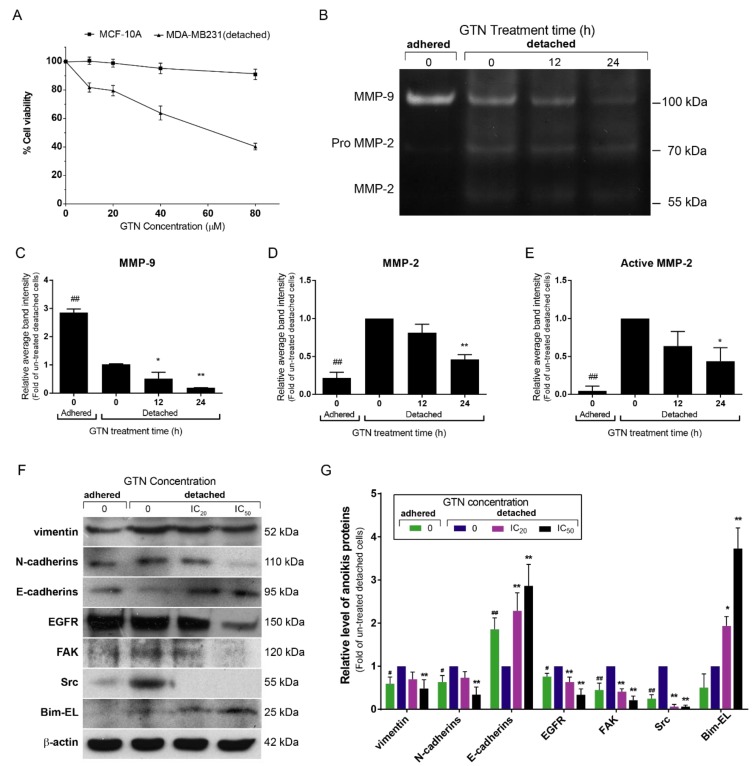
GTN-induced anchorage-independent MDA-MB-231 cell anoikis programmed cell death. The cell viability of detachment MDA-MB-231 cell condition, cultured in a poly-HEMA coated plate were compared to MCF-10A cells as mean ± SD, which was determined by the MTT assay (**A**). Matrix metalloproteinase activity; MMP-2 and MMP-9 were investigated by gelatin zymography (**B**). The bar graphs illustrate the mean ± SD of the average band intensity from three independent experiments of gelatin zymography of MMP-9 (**C**), Pro MMP-2 (**D**), and MMP-2 (**E**). The epithelial-mesenchymal transition (EMT) proteins and the expressions of cell survival molecules, EGFR, FAK, and Src, and the BH3-only protein, Bim-EL, were demonstrated by Western blotting (**F**). The bar graphs illustrated are the mean ± SD of the band density from three independent experiments of Western blotting (**G**). The significance of statistical values from three independent experiments are marked with * *p* < 0.05; ** *p* < 0.01 when compared with the un-treated detached cells and those compared between un-treated groups (adherent cells and detachment cells) were marked with # *p* <0.05; ## *p* < 0.01.

**Figure 6 ijms-20-03953-f006:**
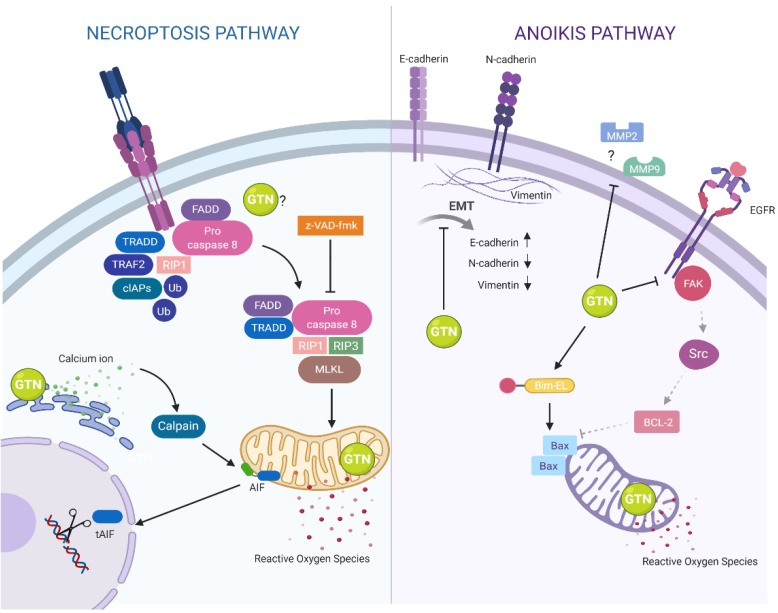
Illustration representing the mechanism of goniothalamin (GTN) co-treated z-VAD-fmk-induced human breast-derived MAD-MB-231 cancer cell necroptosis and anoikis induction in suspension cell culture. GTN induced necroptosis by DNA damage from oxidative stress via an increase of ROS production. GTN directly induced lipid peroxidation at the cell membrane from ROS production, leading to cell membrane rupture. High levels of cytosolic calcium-induced calpain cleave apoptosis-inducing factor (AIF), which translocated to the nucleus for chromatinolysis. z-VAD-fmk also inhibited caspase-8 from executing apoptosis but shifted to necroptosis by the RIP1/RIP3 forming complex with the MLKL protein leading to mitochondria loss of function. In anchorage-independent growth, GTN abrogated detached-culture MDA-MB231 cells and decreased matrix metalloproteinase (MMP-2 and MMP-9) secretion. GTN induced the detached MDA-MB-231 cells to undergo anoikis via epithelial-mesenchymal transition (EMT) by increased E-cadherin protein and decreased N-cadherin and vimentin levels. GTN also inhibited survival signaling through EGFR/FAK/Src and induced expression of the pro-apoptotic protein, Bim-EL, which led to apoptosis-like programmed cell death, anoikis.

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
