# Peer review of "Goniothalamin Induces Necroptosis and Anoikis in Human Invasive Breast Cancer MDA-MB-231 Cells"

_ijms, 2019, doi:10.3390/ijms20163953_

Round 1

Reviewer 1 Report

The current study has reported goniothalamin can induce necroptosis and anoikis of human invasive breast cancer MDA-MB-231 cells in a caspase-independent manner, which is novelty. And the molecular signaling pathways have also been measured, which are classical and seem less novelty.

There are some concerns that need to be addressed.

Major issues: 

According to Figure 1A, IC50 are 40-60 μM in the presence or absence of z-VAD-fmk, please check “33 μM” and “37 μM” in Result 2.1, line 4.

Like Figure 1A and 1B, cell morphology of MDA-MB-231 cells with GTN treatment alone should be observed.

“pan-caspase inhibitor shifted the mode of cell death induced by GTN to be necrosis from apoptosis” in Result 2.1: whether GTN can induce apoptosis of MDA-MB-231 cell lines should be measured, or at least, relevant study should be cited.

In result 2,4, mRNA expression levels of rip1, rip3 and mlkl have been measured, but only protein expression level of RIP3 has been measured, what about the other two?

In result 2.4, calpain activity was significantly increased in GTN w/ z-VAD-fmk group when compared to control and GTN treatment alone, while in Figure 2B legend, ** means the significance of statistical values compared to control group (without treatment). Please unify the statement.

In Figure 3C, comparisons between GTN w/ z-VAD-fmk group and GTN wo/ z-VAD-fmk group under IC20 and IC50 should be added.

Figure 4 legend is incomplete. Please complete it.

In Figure 5, EMT markers have been measured, which are involved in process of cancer cells invasion and migration. Therefore, the expression of MMPs should also be measured.

The last sentence of Figure 6 legend seems incomplete, please check.

The discussion about Result 2.1 is deficient, please complete it.

Minor issues: 

Please check the sentence structure of conclusion part in abstract: “Anoikis-resistant cell death was induced in MDA-MB-231 cells by GTN……”.

Page 5, 2.3. title: “pan-caspase inhibitor” could be modified to “z-VAD-fmk”.

Author Response

The current study has reported goniothalamin can induce necroptosis and anoikis of human invasive breast cancer MDA-MB-231 cells in a caspase-independent manner, which is novelty. And the molecular signaling pathways have also been measured, which are classical and seem less novelty.

-          Thank you, the Reviewer 1 for your opinions and suggestions

There are some concerns that need to be addressed.

Major issues:

1. According to Figure 1A, IC50 are 40-60 μM in the presence or absence of z-VAD-fmk, please check “33 μM” and “37 μM” in Result 2.1, line 4.

- We re-check and corrected the inhibitory concentration at 50 percent by using a non-linear curve fit method (Figure 1A). The analysis result was shown as below and the values were revised to the Result 2.1 (line 75-77). The inhibitory concentration at 50 percent (IC50) was 33.82 µM compared to 44.65 µM without z-VAD-fmk, but GTN was less cytotoxic to MCF-10A which is a non-tumorigenic human breast epithelial cell line, with IC50 of > 80 µM. 

log(inhibitor) vs. normalized response --   Variable slope

Best-fit values

GTN

GTN+ z-VAD-fmk

LogIC50

1.650

1.529

HillSlope

-1.142

-0.9859

IC50

44.65

33.82

Std. Error

LogIC50

0.03624

0.03796

HillSlope

0.1353

0.1157

95% Confidence Intervals

LogIC50

1.572 to 1.728

1.447 to 1.611

HillSlope

-1.434 to -0.8497

-1.236 to -0.7360

IC50

37.29 to 53.47

28.00 to 40.84

Goodness of Fit

R square

0.9369

0.9405

2. Like Figure 1A and 1B, cell morphology of MDA-MB-231 cells with GTN treatment alone should be observed.

- Under the subtopic: 2.1. Cytotoxic and necroptotic effects on MDA-MB-231 cells. we added two TEM results (Figure 1); GTN and z-VAD-fmk treated alone to confirm apoptosis and non-cytotoxic effect, respectively. (line 80-88) and described in the Results:

2.1 Cytotoxic and necroptotic effects on MDA-MB-231 cells…………………..“Transmission electron micrographs (TEM) revealed cell morphology of untreated-MDA-MB231 cells which was intact and sealed cell membrane (Figure. 1C, no arrowhead) as same as z-VAD-fmk treated cells which are not toxic towards MDA-MB-231 cells (Figure. 1F). GTN and z-VAD-fmk co-treatment demonstrated MDA-MB-231 cell loss of their cell membrane integrity (red arrowhead as in Figure. 1D) as well as the positive control hydrogen peroxide (Figure. 1E). In contrast, GTN-treated (alone) cells revealed cell shrinkage and DNA condensation and membrane blebbing (blue arrow) which was apoptosis feature (Figure. 1G). When combined with our previous study it revealed that GTN alone induced MDA-MB231 cell apoptosis [19]. Hence, z-VAD-fmk may shift the mode of cell death induced by GTN to be necrosis rather than apoptosis, which is mediated through caspases, therefore apoptosis was inhibited by pan-caspase inhibitor.”

3. “pan-caspase inhibitor shifted the mode of cell death induced by GTN to be necrosis from apoptosis” in Result 2.1: whether GTN can induce apoptosis of MDA-MB-231 cell lines should be measured, or at least, relevant study should be cited.

- Thank you for your suggestion, we cited our recent study which revealed goniothalamin induced MDA-MB231 apoptosis. (line 86-87) We cited our previous study that goniothalamin (GTN) induced MDA-MB-231 cell apoptosis.When combined with our previous study it revealed that GTN alone induced MDA-MB231 cell apoptosis [19].

4. In result 2,4, mRNA expression levels of rip1, rip3 and mlkl have been measured, but only protein expression level of RIP3 has been measured, what about the other two?

- Thank you for your concern, we discussed the necroptosis regulators and the molecules specific of machinery in the Discussion section. In 4th Paragraph of Discussion (line 264-270): RIP1 is an essential protein in the death-inducing signaling complex (DISC) that is involved in death-receptor-initiated cell death, apoptosis, and necroptosis [45]. The receptor-specific necroptosis signaling, RIP3 pathway is a crucial upstream kinase which mediates downstream signaling such as MLKL [46]. Here, we also demonstrate the potential of GTN to induce the breast cancer-derived cell line MDA-MB-231 to undergo necroptosis via the RIP3 pathway with increased RIP3 protein expression, whereas rip1, rip3, and mlkl gene expressions also significantly enhanced, finally leading to necroptosis as illustrated in Figure 6.

5. In result 2.4, calpain activity was significantly increased in GTN w/ z-VAD-fmk group when compared to control and GTN treatment alone, while in Figure 2B legend, ** means the significance of statistical values compared to control group (without treatment). Please unify the statement.

-We unified the significant symbols including in Figure2 (line 116-118)

·         Star (*), when compared with control (un-treated group)

·         Sharp (#), when compared between GTN alone with GTN+z-VAD-fmk

6. In Figure 3C, comparisons between GTN w/ z-VAD-fmk group and GTN wo/ z-VAD-fmk group under IC20 and IC50 should be added.

- we added the statistical comparison between GTN w/ z-VAD-fmk group and GTN wo/ z-VAD-fmk group under IC20 and IC50 as the Reviewer 1 mentioned. (Figure 3C)

7. Figure 4 legend is incomplete. Please complete it.

- we completed Figure 4 legend as the reviewer’s recommend. (line 167-169): The data were obtained from 3 independent experiments in triplicate, the significance of statistical values compared to the z-VAD-fmk-treated alone group are marked with *, p<0.05; **, p<0.01 and compared to GTN alone group are marked with ##, p<0.01

8. In Figure 5, EMT markers have been measured, which are involved in process of cancer cells invasion and migration. Therefore, the expression of MMPs should also be measured.

- we agree with the reviewer’s suggestion, we added MMP secretion result to complete the effect of GTN on invasion potential. (Figure 5) Also, the results were described in Results :

2.5. Goniothalamin-reversed anoikis resistance…………….

In 2nd paragraph (line 183-190)

9. The last sentence of Figure 6 legend seems incomplete, please check.

- The sentence was revised to improve the flow. (line 297-303)The sentence was added at the final of Figure 6. As follows: In anchorage-independent growth, GTN abrogated detached-culture MDA-MB231 cells and decreased matrix metalloproteinases (MMP-2 and MMP-9) secretion. GTN induced the detached MDA-MB-231 cells to undergo anoikis via epithelial-mesenchymal transition (EMT) by increased E-cadherin protein and decreased N-cadherin and vimentin levels. GTN also inhibited survival signaling through EGFR/FAK/Src and induced expression of pro-apoptotic protein, Bim-EL which leading to apoptosis-like programmed cell death, anoikis.

10. The discussion about Result 2.1 is deficient, please complete it.

- thank you for your suggestion, we discuss more detail about programmed cell death shifting. (line 218-243): in the 1st paragraph of the Discussion as follows:

GTN has been reported to induce apoptosis cell death in many cancer cell types [1]. The effect of GTN on cancer cell apoptosis, such as DNA damage and cell cycle arrest, is mediated via both intrinsic and extrinsic apoptosis pathways [4, 28, 29]. Necrosis morphology has previously been found in MCF-7 breast cancer cells [7]. Our previous study showed that GTN induced MDA-MB231 cells to undergo apoptosis. Apoptosis induction in GTN-treated MDA-MB231 mediated endoplasmic reticulum and oxidative stresses. Intrinsic and extrinsic apoptosis were revealed, which altered apoptosis protein expressions. GTN-treated MDA-MB231 cells also arrested at G2/M cell cycle phase [19]. Interestingly, a recent study reported that co-treatment with z-VAD-fmk (a pan-caspase inhibitor) inhibits apoptosis by completely inhibiting caspases, resulting in a shift from apoptosis to necroptosis in TNF-treated L929 cells [30]. Cisplatin also shows the potential of necroptosis induction when co-treated with pan-caspase inhibitor on gastric cancer [31]. Reports regarding other natural products or extracts, namely green tea extract, shikonin, and columbianadin, demonstrate the results of necroptosis induction, which is call “regulated necrosis” [32-34]. The necroptotic cell death can be combined or used alone as more probability and benefits for cancer therapy when considering the mechanism of action at the molecular level. However, investigation of GTN on other programmed cell death inductions and the mechanisms involved remains unclear. Necrosis morphology has previously been found in MCF-7 breast cancer cells [7] by TEM.

Here, we revealed the different modes of cell death in GTN-treated MDA-MB-231 cells treated with z-VAD-fmk. Ultra-fine structure of GTN co-treated with z-VAD-fmk cells exhibited the large vacuoles, ruptured cell membrane and released cellular organelles, which are as same as hydrogen peroxide-induced necrosis positive control (Figure. 1). This indicated that GTN induced cell death in different modes by shifting from apoptosis to necrosis when co-treated with the pan-caspase inhibitor. For cytotoxicity of normal cells, goniothalamin was less toxic to normal cell types compared to cancer cell types [35-37], this study has also added the cytotoxicity data of normal mammalian gland epithelial MCF-10A cell line, of which GTN was less cytotoxic as ten percent compared to control cells (Figure. 1). This indicated that goniothalamin is a natural compound, which possesses the specific toxicity towards cancer cells.

Minor issues:

1. Please check the sentence structure of conclusion part in abstract: “Anoikis-resistant cell death was induced in MDA-MB-231 cells by GTN……”.

- we checked and revised the sentence as the Reviewer 1’s suggestion. (page 1, line 22-24)

GTN induced MDA-MB-231 cells to undergo anoikis via reversed epithelial-mesenchymal transition (EMT) protein expressions, inhibited EGFR/FAK/Src survival signaling pathway, and decreased matrix metalloproteinase secretion.

2. Page 5, 2.3. title: “pan-caspase inhibitor” could be modified to “z-VAD-fmk”.

- we agree and change the title from “pan-capase inhibitor” to “z-VAD-fmk” (line 118)

In the subtitle: 2.3 we changed the subtitle to be: 2.3. Induction of high cellular ROS production and increased cytosolic Calcium level by GTN co-treated with z-VAD-fmk in MDA-MB-231 cells

Reviewer 2 Report

Authors pointed out an interesting and challenging topic. Making the drug-resistant cell sensitive would be very applicable in clinical trial. However, the authors strategy and the design of experiments might consider as inadequate and preliminary. 

To find out the mechanism that is involved in cancer-cell vulnerability, authors touched different pathways but did not go deeply into it and thus the question is still open: what was the mechanism involved in GTN-induced apoptosis? Authors might take a more comprehensive strategy to address this interesting question, combine the experiments with other inhibitors. 

A FACS analysis to analyse the cell cycle and to monitor the apoptosis status might help too. If there is no finance limitation including other cell lines (invasive and non-invasive as control) and adding negative and positive controls would be as needed to evaluate the data in correct way.

As authors are trying to answer the metastatic potential of the cell, including migration assay or similar experiments is needed.

Author Response

Authors pointed out an interesting and challenging topic. Making the drug-resistant cell sensitive would be very applicable in clinical trial. However, the authors strategy and the design of experiments might consider as inadequate and preliminary. 

-          Thank you to the Reviewer 2 for the comments, as the results on this manuscript study was about other programmed cell deaths whether GTN or GTN plus z-VAD-fmk could reverse apoptosis resistance in human breast cancer cells to die via other kinds of cell death. This study was in vitro study and will lead to more comprehensive studies. The further study in cell signaling molecules involved and in vivo or even clinical trial are needed before application in human patients.

To find out the mechanism that is involved in cancer-cell vulnerability, authors touched different pathways but did not go deeply into it and thus the question is still open: what was the mechanism involved in GTN-induced apoptosis? Authors might take a more comprehensive strategy to address this interesting question, combine the experiments with other inhibitors. 

-          We also concerned about other signaling pathways of Goniothalamin-induced cell death. However, we have reported the apoptosis induction and its involved mechanisms, which we cited in the Discussion section. (line 218-225):

In the 1st paragraph of Discussion, the first two sentences referred to GTN-induced apoptosis and the signaling pathway in MDA-MB-231 apoptotic cells as cited in References 1, 4, 28, and 29. Moreover, we reported and focused in our previous apoptosis pathway induced by GTN in Ref. 19.

A FACS analysis to analyse the cell cycle and to monitor the apoptosis status might help too. If there is no finance limitation including other cell lines (invasive and non-invasive as control) and adding negative and positive controls would be as needed to evaluate the data in correct way.

-          We previously reported GTN induced MDA-MB231 cell apoptosis via oxidative and endoplasmic reticulum (ER) stresses. The study was also revealed cell cycle and apoptosis analysis. In addition, we published the cytotoxic and apoptosis induction effect of goniothalamin in MCF7, HeLa and HepG2 cell lines which induced in different pathways of apoptosis in Ref. 19.

-          We added the details of our previous study in Discussion for more comprehensive signaling and the effect of GTN-induced apoptosis on human cancer cells Reference 19. We mentioned in Discussion section to clarify the effect of GTN on other types of cell death, i.e.,  necrosis, or regulated necrosis (necroptosis) with the typical morphology. The effect of z-VAD-fmk (pan-caspase inhibitor) was to inhibit caspase 8 and shifedt apoptosis to necroptosis in the present study. (line 225-230)

As authors are trying to answer the metastatic potential of the cell, including migration assay or similar experiments is needed.

-          We demonstrated MMP-2 and MMP-9 secretion by gelatin zymography which represents the invasive potential of MDA-MB231 (Figure 5). Also, there are reports of GTN in migration inhibition which were cited in the Discussion. (line 274-278)

Reviewer 3 Report

Goniothalamin ( GTN) has been previously reported to inhibit proliferation, cause cell cycle arrest, and induce apoptosis in  several cancerous cell lines.

GTN inhibited proliferation and induced necrosis of MCF-7 cells through mechanisms that are not completely understood.

In MDA-MB-231 cells, GTN caused a G2/M cell cycle arrest, ROS production and DNA damage, which lead to cell death.

The authors hypothesize that GTN acts on pathways of Necroptosis and sensitizes cancer cells to anoikis that abrogates anoikis-resistant pathways in cancer cells.

In the current study, the authors use MDA-MB-231 cells as a model of anchorage-independent invasive cancer cells to study the anti-cancer mechanisms of GTN.

The manuscript has well stated objectives. However, the study depended on experiments conducted on only one invasive anchorage-independent cancerous cell line. The authors should evaluate their results in other invasive anchorage-independent cancerous cell lines. Also, MDA-MB-231 cells are a breast cancer cell line. Do the obtained results apply to all cancer types or are they specific to breast cancer. The authors should evaluate their results using cancer cell lines from tissues other than breast.

The results lack data from proper controls:

Figure 1A

There is no advantage of adding the zVAD inhibitor. The % viability is almost similar between the curves of with and with-out zVAD. At 40 μM and  beyond, statistical analysis showed no difference between with and without zVAD.

Figure 1 B

Should show data for cells treated with GTN alone.

Figure 1 B

It is preferable to draw the figure as: % of propidium iodide positive cell population (Fold of un-treated control). Where the untreated control is not treated with either zVAD or GTN.

Figure 1

Authors need to show TEM of cells treated with GTN alone and cells treated with zVAD alone otherwise the conclusion “pan- caspase inhibitor shifted the mode of cell death induced by GTN to be necrosis from apoptosis”

Figure 2A

Should show data for control untreated cells, cells treated with zVAD alone, and cells treated with GTN alone.

Figure 2B

need to show data for zVAD alone.

Figure 3, The control data are actually not control. The control should be untreated cells, or cells treated with vehicle. While the controls displayed in Figure 3 are either zVAD with not GTN or zVAD with NAC. The authors need to redraw the figures with fold change corresponding to untreated cells also they need to show zVAD alone data in all figure panels, and probably NAC alone data, or mention in the text that NAC alone had no effect.

Minor points:

Line 58-59 from “Apoptosis-related” till end anoikis need reference.

Line 78-79 “the PBMCs as normal control cells, with IC50  of > 80 μM”  from Figure 1A, even 80 μM of GTN killed only about 8% of PBMCs (from 100% to around 92%). I think the authors should use a better positive control than PBMCs (which are apparently resistant to GTN induced cell death) or clarify in the text why they chose PBMCs.

Figure 1A, the significance symbols should be above the blue curve not the green curve.

The manuscript need thorough English editing:

Line 83-84 “necrosis cells as positive hydrogen peroxide- treated cells ( Figure.  1E)”  better to use: “as well as the positive control hydrogen peroxide”

Line 85 “to be necrosis from apoptosis”  probably change to; “to be necrosis rather than apoptosis”

Line 85 “which mediated through caspase activity [18].” Change to: “which is mediated”.

Line 99 “or without treatment, it did not induce “ change to: ““or without treatment, showed that it did not induce”

Line 111-112 “From 3 independent experiments in triplicate,” add “Data are obtained from”

Line 116 “previous authors” change to: for example “previous studies”

Line 118 “ involved with” remove with

Author Response

Goniothalamin (GTN) has been previously reported to inhibit proliferation, cause cell cycle arrest, and induce apoptosis in several cancerous cell lines.

GTN inhibited proliferation and induced necrosis of MCF-7 cells through mechanisms that are not completely understood.

In MDA-MB-231 cells, GTN caused a G2/M cell cycle arrest, ROS production and DNA damage, which lead to cell death.

The authors hypothesize that GTN acts on pathways of Necroptosis and sensitizes cancer cells to anoikis that abrogates anoikis-resistant pathways in cancer cells.

In the current study, the authors use MDA-MB-231 cells as a model of anchorage-independent invasive cancer cells to study the anti-cancer mechanisms of GTN.

The manuscript has well stated objectives. However, the study depended on experiments conducted on only one invasive anchorage-independent cancerous cell line. The authors should evaluate their results in other invasive anchorage-independent cancerous cell lines. Also, MDA-MB-231 cells are a breast cancer cell line. Do the obtained results apply to all cancer types or are they specific to breast cancer. The authors should evaluate their results using cancer cell lines from tissues other than breast.

-          The objective of this research is to study whether goniothalamin induced MDA-MB-231 to undergo non-apoptosis cell death. The first one, we disrupt the apoptosis by using a  pan-caspases inhibitor which was used to study non-apoptosis in anchorage growth. The result revealed that goniothalamin induced MDA-MB231 cells to necroptosis when caspase enzymes has been abrogated. The second, we cultured MDA-MB231 cells in anchorage-independent growth to simulate or mimic the invasion and migration stage. This result showed that goniothalamin also induced the cancer cells to anoikis.

-          This manuscript will fulfill the effect of goniothalmin-induced multi-program cell deaths which depend on the micro-environment or milieu in culture system, including apoptosis in normal culture, necroptosis in all caspase suppression and anoikis in suspension culture. This leads to conclusion that goniothalamin could be used as chemo drug in different stages of cancer, adherent (anchorage dependent, non-invasive), anchorage independent (invasive), which was designed in an in vitro such as in breast cancer.

-          We have cited other studies that involved goniothlamin-induced cell death in other types of cancer. The discussion was added for other types of cancer, (line 218-220) and other compounds or natural products in the 1st paragraph of Discussion.

-          : as Other cells types are cited in the 1st paragraph of Introduction follows:            in Ref. 2, 3

-          GTN inhibits the proliferation of HCG-27, MCF7, PAN C-1, and HeLa cancerous cell-lines and noncancerous fibroblast NIH3T3 cell lines cultured in vitro [1], and also induces apoptosis in Jurkat T-cells and human promyelocytic leukemia HL-60 cells via caspase-3, -7 and poly(ADP-ribose)polymerase-1 (PARP-1) cleavage [2,3].   

The results lack data from proper controls:

Figure 1A

There is no advantage of adding the zVAD inhibitor. The % viability is almost similar between the curves of with and with-out zVAD. At 40 μM and beyond, statistical analysis showed no difference between with and without zVAD.

-          The caspase inhibitor in the first figure did not aim for synergistic effect when co-treated with goniothalamin, instead we used to abrogate caspase activity in cancer cell to study caspase-independent non-apoptosis induction of Goniothalamin, which is necroptosis. From the Figure 1A, goniothalamin still useful in term of caspase malfunction in cancer cells (inhibition of caspase activity) which can be found in aggressive cancer cell type; highly anti-apoptosis gene expression or malfunction

-          We discussed more details in Discussion section (line 223-230): showing that necroptosis might be the targeted cell death in many kinds of cancer to target at the signaling molecules. The other natural products or other active compounds can also induce necroptosis and apoptosis depending on doses used.

-          GTN has been reported to induce apoptosis cell death in many cancer cell types [1]. Cisplatin also shows the potential of necroptosis induction when co-treated with pan-caspase inhibitor on gastric cancer [31]. Reports regarding other natural products or extracts, namely green tea extract, shikonin, and columbianadin, demonstrate the results of necroptosis induction, which is call “regulated necrosis” [32-34]. The necroptotic cell death can be combined or used alone as more probability and benefits for cancer therapy when considering the mechanism of action at the molecular level.

Figure 1 B

Should show data for cells treated with GTN alone.

-          We already compared GTN co-treated with z-VAD-fmk (black bar) with GTN alone (gray bar) and normalized with untreated cells . However, we change the bar graph label to GTN and GTN+z-VAD-fmk instead of /wo z-VAD-fmk and /w z-VAD-fmk, respectively to prevent misunderstanding.  (Figure 1 B)

Figure 1 B

It is preferable to draw the figure as: % of propidium iodide positive cell population (Fold of un-treated control). Where the untreated control is not treated with either zVAD or GTN.

-          Thank you for your concern, we re-calculated the PI positive cell population by normalized with un-treated cells as the Reviewer mentioned (represent as fold of un-treated control)

Figure 1

Authors need to show TEM of cells treated with GTN alone and cells treated with zVAD alone otherwise the conclusion “pan- caspase inhibitor shifted the mode of cell death induced by GTN to be necrosis from apoptosis”

-          Thank you for your suggestion, as well as the Reviewer 1, we added two TEM results; GTN and z-VAD-fmk treated alone to confirm apoptosis and non-cytotoxic effect, respectively

Figure 2A

Should show data for control untreated cells, cells treated with zVAD alone, and cells treated with GTN alone.

-          As your suggestion, we re-plotted caspase activity results by showing those activities in un-treated cells, GTN and z-VAD-fmk alone and GTN+z-VAD-fmk treatment, respectively.

Figure 2B

need to show data for zVAD alone.

-          Thank you for your concern, we added the calpain activity result of z-VAD-fmk treated cells to cut-off the effect of z-VAD-fmk in calpain induction. 

Figure 3, The control data are actually not control. The control should be untreated cells, or cells treated with vehicle. While the controls displayed in Figure 3 are either zVAD with not GTN or zVAD with NAC. The authors need to redraw the figures with fold change corresponding to untreated cells also they need to show zVAD alone data in all figure panels, and probably NAC alone data, or mention in the text that NAC alone had no effect.

-          We apologize for the unclear result figure and we understand your concern. We normalized the data with un-treated group. It’s our missing that labeled fluorescence intensity mean as fold of control vice fold of un-treated.

-          In the CON bar group,

·         CON- GTN w/ z-VAD-fmk (Black bar) = z-VAD-fmk alone normalized with untreated group

·         CON- GTN w/ z-VAD-fmk+NAC (Black patterned bar) = z-VAD-fmk and NAC treatment normalized with untreated group

·         CON- GTN (Gray bar) = non-treatment normalized itself

·         CON- GTN+NAC (Gray patterned bar) = NAC alone normalized with untreated group

So, we changed figure labels to clarify the treatment conditions and were described in the Result context. 

Minor points:

Line 58-59 from “Apoptosis-related” till end anoikis need reference.

-          We added reference to support those sentences as the reviewer suggestion. (line 56-57F) Ref. 15

Line 78-79 “the PBMCs as normal control cells, with IC50 of > 80 μM” from Figure 1A, even 80 μM of GTN killed only about 8% of PBMCs (from 100% to around 92%). I think the authors should use a better positive control than PBMCs (which are apparently resistant to GTN induced cell death) or clarify in the text why they chose PBMCs.

-          Thank you for your notice, we agree to change a normal cell control from PBMC to MCF-10A which is a non-tumorigenic breast epithelial cell line. The result revealed better resistance to goniothalamin than PBMC which corresponding to the Reviewer 3 suggestion. (Figure1)  

Figure 1A, the significance symbols should be above the blue curve not the green curve.

-          We changed the significance symbols as your suggestion. (Figure1)

The manuscript need thorough English editing:

Line 83-84 “necrosis cells as positive hydrogen peroxide- treated cells ( Figure.  1E)”  better to use: “as well as the positive control hydrogen peroxide”

-          We agreed and changed as the reviewer’s recommended (line 83-84)

Line 85 “to be necrosis from apoptosis” probably change to; “to be necrosis rather than apoptosis”

-          We agreed and changed as the reviewer’s recommended (line 87, 88)

Line 85 “which mediated through caspase activity [18].” Change to: “which is mediated”.

-          We agreed and changed as the reviewer’s recommended (line 88)

Line 99 “or without treatment, it did not induce “ change to: ““or without treatment, showed that it did not induce”

-          We agreed and changed as the reviewer’s recommended (line 103-104)

Line 111-112 “From 3 independent experiments in triplicate,” add “Data are obtained from”

-          We agreed and added as the reviewer’s recommended (line 116)

Line 116 “previous authors” change to: for example “previous studies”

-          We agreed and changed as the reviewer’s recommended (line 121)

Line 118 “involved with” remove with

-          We agreed and removeed as the reviewer’s recommended (line 123)

Round 2

Reviewer 1 Report

The revised manuscript has been modified in accordance with the comments.

Reviewer 2 Report

---